# Robust Pavement Modulus Prediction Using Time-Structured Deep Models and Perturbation-Based Evaluation on FWD Data

**DOI:** 10.3390/s25175222

**Published:** 2025-08-22

**Authors:** Xinyu Guo, Yue Chen, Nan Sun

**Affiliations:** 1Faculty of Information Science and Technology, University Kembangan Malaysia, Bangi 43600, Malaysia; guoxy395397977@gmail.com; 2School of Engineering and Technology, The University of New South Wales, Canberra, ACT 2600, Australia; yue.chen4@unsw.edu.au; 3School of Systems & Computing, The University of New South Wales, Canberra, ACT 2600, Australia

**Keywords:** pavement modulus prediction, FWD, deep learning, sequence modeling, perturbation robustness, ResRNN

## Abstract

The accurate prediction of the pavement structural modulus is crucial for maintenance planning and life-cycle assessment. While recent deep learning models have improved predictive accuracy using Falling Weight Deflectometer data, challenges remain in effectively structuring time-series inputs and ensuring robustness against noise measurement. This paper presents an integrated framework that combines systematic time-step modeling with perturbation-based robustness evaluation. Five distinct input sequencing strategies (Plan A through Plan E) were developed to investigate the impact of temporal structure on model performance. A hybrid Wide & Deep ResRNN architecture incorporating SimpleRNN, GRU, and LSTM components was designed to jointly predict four-layer moduli. To simulate real-world sensor uncertainty, Gaussian noise with ±3% variance was injected into inputs, allowing the Monte-Carlo-style estimation of confidence intervals. Experimental results revealed that time-step design plays a critical role in both prediction accuracy and robustness, with Plan D consistently achieving the best balance between accuracy and stability. These findings offer a practical and generalizable approach for deploying deep sequence models in pavement modulus prediction tasks, particularly under uncertain field conditions.

## 1. Introduction

The accurate estimation of pavement layer moduli is essential for infrastructure evaluation, life-cycle cost analysis, and maintenance planning [1]. Among non-destructive testing methods, the Falling Weight Deflectometer (FWD) is widely used due to its ability to simulate dynamic wheel loading and capture multi-sensor deflection responses, which reflect the layered mechanical behavior of pavements [2,3]. A schematic diagram of the FWD test is shown in Figure 1. With the increasing availability of FWD datasets from platforms like the Long-Term Pavement Performance (LTPP) database, data-driven methods have emerged as efficient alternatives to traditional back-calculation [4,5,6,7,8]. Early studies mainly used shallow regressors while recent work has adopted recurrent neural networks (RNNs), GRUs, and LSTMs [9,10,11,12,13,14,15,16,17,18,19,20] to model nonlinear relationships between surface deflections and subsurface moduli. However, FWD data inherently exhibit spatiotemporal structures, with sequential deflections modulated by temperature gradients, material heterogeneity, and site conditions [13,14,15,16]. Capturing these dynamics requires input representations that align with physical response mechanisms, not just deeper model architectures. This study investigated whether physically informed input structuring could enhance predictive accuracy and robustness in FWD-based modulus prediction, bridging pavement mechanics with data-centric learning.

Despite the encouraging performance of deep learning models in pavement modulus prediction, existing studies share several methodological shortcomings that limit both their accuracy and practical deployment. A central issue lies in the treatment of input features: many models flatten sequential deflection signals, thermal measurements, and material attributes into static vectors, stripping away the temporal and spatial structure inherent to FWD data [6]. This undermines the advantages of sequence-based architectures like RNNs and GRUs, which are designed to capture ordered dependencies [7]. Moreover, most models are developed under idealized assumptions of clean, stationary data, overlooking the sensor noise, environmental variability, and operational inconsistencies commonly encountered in the field [8,9]. As a result, their robustness under real-world perturbations remains uncertain. Compounding these issues, many approaches neglect the functional roles of different features—such as distinguishing between stimuli (e.g., load, temperature) and system responses (e.g., deflections)—forcing models to relearn physical relationships that are already well-understood [10]. Without a meaningful grouping and semantic structuring of inputs, the learning process becomes less efficient and more prone to overfitting.

Although deep learning has gained traction in modeling FWD-based pavement layer moduli, several critical gaps remain unresolved. Most prior studies relied on static, flattened input formats that overlooked the sequential nature of deflection propagation and the layered structure of pavements, limiting the effectiveness of temporal models like RNNs, GRUs, and LSTMs in capturing interlayer dependencies and causal dynamics [11,12,13]. Compounding this, models are typically developed and validated under idealized, noise-free conditions, with little attention to the impact of sensor drift, calibration variability, and environmental noise on prediction stability [14,15]. Furthermore, there is limited integration between input design strategies and robustness evaluation: few works have assessed how temporally or functionally structured inputs affect model learning under perturbation, and even fewer propose a unified framework for such testing [16,18]. Instead, much of the literature emphasizes architectural complexity—deeper layers, attention mechanisms—without addressing the foundational role of physically meaningful feature structuring. Without causal ordering or semantic grouping, even sophisticated networks may yield brittle or uninterpretable outputs in practice. These limitations reveal a dual gap: the lack of input representations grounded in physical behavior and the lack of robustness-aware validation. Addressing both is essential for transitioning deep learning models from theoretical success to reliable field deployment in noisy, real-world environments [17,19,20,21].

The recurring limitations in prior work highlight that the core challenge in applying deep learning to FWD-based pavement analysis lies less in model architecture than in the design of input representation. We propose shifting from an algorithm-centric paradigm to an input-centric perspective, where physically grounded structuring—such as in temporal ordering, semantic grouping, and causal separation—forms the foundation of model learning [22]. To test this hypothesis, we introduce a family of time-stepped input configurations (Plans A–E) that embed domain knowledge directly into the sequence structure. For example, Plan A organizes deflection and temperature readings by sensor distance from the load plate, capturing stress wave propagation patterns. Plan D further segments input into six behavioral stages, distinguishing between load stimuli, thermal gradients, near and far deflection zones, and structural layer attributes. To assess robustness, we inject controlled Gaussian perturbations into key variables, simulating field noise and enabling stochastic performance comparisons across input plans [23]. By explicitly separating causal inputs (e.g., load, temperature) from reactive outputs (e.g., deflections), this framework aims to enhance learning interpretability and maintain model stability under uncertainty.

We present a novel framework that (i) encodes pavement mechanics through domain-informed input sequencing (Plans A–E, with causal Plan D) and (ii) rigorously evaluates robustness via controlled noise perturbations and uncertainty metrics. First, instead of treating the input features as unordered or flattened vectors, we explore multiple sequence construction strategies rooted in pavement mechanics. Among these, Plan D organizes inputs by causally grouping loading conditions, temperature profiles, deflection signals, and structural attributes, allowing the model to better reflect the layered response process observed during FWD testing. Second, we incorporate a noise-injection framework that simulates realistic field disturbances, such as sensor drift and environmental fluctuations, to evaluate model stability under uncertainty. This dual emphasis on physically structured inputs and robustness assessment distinguishes our approach from prior deep learning applications, where architectural depth was often prioritized over domain-informed feature representation and real-world reliability.

This paper presents a physically informed, robust-oriented framework for FWD-based pavement modulus prediction. Our main contributions include the following:Structured Input Sequencing: We propose five input plans (Plan A–E) that encode deflection propagation and material layering into time-step sequences [5,16,17,24].Perturbation-Based Robustness Testing: We introduce Gaussian noise to simulate field variability and assess model stability under repeated disturbances [25,26,27,28].Semantic Feature Grouping: We separate causal stimuli from system responses to enhance interpretability and reduce learning complexity [29,30,31].Dual-Criterion Evaluation: We establish a practical model selection strategy that jointly considers prediction accuracy and robustness in noisy environments [32].

Together, these contributions form a generalizable framework for integrating domain logic into deep learning workflows, with applications in pavement evaluation and time-series modeling under uncertainty.

Compared to conventional pavement modulus estimation techniques, particularly the widely adopted back-calculation methods, the data-driven approach proposed in this study offers several distinct advantages. Back-calculation involves an iterative optimization process to estimate moduli that minimizes the deviation between measured and simulated deflections. Despite its longstanding use in engineering practice, this method is often criticized for producing non-unique solutions, being sensitive to measurement noise, and heavily relying on initial assumptions such as layer thicknesses and Poisson’s ratios. In some cases, it may suffer from convergence issues. Moreover, back-calculation procedures typically require substantial expert supervision and are less amenable to automation or real-time implementation [33,34,35,36,37].

In contrast, the machine learning and deep learning models developed in this study are capable of directly mapping sensor inputs to modulus values without requiring initial assumptions. These models demonstrate greater robustness under perturbation, allow for rapid inference, and scale well with larger datasets. Nonetheless, learning-based approaches also face challenges, including dependency on representative training data, the need for interpretability, and potential overfitting risks. Taken together, these considerations highlight a trade-off between physical interpretability and predictive scalability, underscoring the need for hybrid approaches and expanded validation under field conditions.

## 2. Background

### 2.1. Modulus Prediction Methods

Traditional pavement evaluation methods often rely on back-calculation techniques, which iteratively estimate the elastic modulus of each pavement layer from surface deflection measurements [38]. While widely adopted, these techniques suffer from high computational costs, sensitivity to initial assumptions, and non-uniqueness in their solutions, particularly when noise or material heterogeneity is present.

To overcome these limitations, machine learning (ML) models such as Support Vector Regression (SVR), Random Forests (RFs), and Gradient Boosting Machines (GBMs) have been introduced, offering enhanced computational efficiency and better generalization to unseen data. These models reduce reliance on inverse mechanics and can learn empirical mappings directly from deflection data to modulus values [5,8,39].

More recently, deep learning (DL) techniques, especially feedforward Artificial Neural Networks (ANNs), have further improved predictive performance by capturing complex nonlinear relationships within the data. However, their effectiveness depends critically on how well the underlying physics of the pavement system is encoded in the input structure, a consideration often overlooked in model design [40,41].

### 2.2. Time-Series Deep Learning in Structural Prediction

Recurrent neural networks (RNNs), long short-term memory networks (LSTMs), and gated recurrent units (GRUs) have become standard tools for modeling sequential data in domains such as energy forecasting, asset degradation, and structural health monitoring [42,43]. These models are explicitly designed to capture transitions and dependencies across time steps or spatial gradients, offering strong representational power when inputs follow a meaningful temporal or spatial order.

In the context of pavement modulus prediction, however, these architectures have been inconsistently applied. Most prior studies have treated Falling Weight Deflectometer (FWD) inputs, such as deflection signals, temperature readings, and layer types, as flat and unordered feature vectors. This approach disregards the spatial propagation of stress waves, the staged structural response of layered pavements, and the causal sequencing of physical events during FWD testing. Consequently, the potential of sequence-based models is left unrealized as their temporal learning mechanisms are never fully activated [8,11].

### 2.3. Stability and Robustness Assessment

In mission-critical applications, model robustness under noisy or perturbed inputs is a foundational requirement. To this end, various disciplines have adopted methods such as Monte Carlo dropout, confidence interval estimation, and ensemble averaging to evaluate prediction stability and quantify uncertainty [44,45]. Surprisingly, such practices have seen limited adoption in pavement engineering, despite the inherently noisy nature of field data. Variations in sensor calibration, environmental conditions, equipment wear, and operator procedure frequently introduce fluctuations in FWD measurements. Yet, most studies continue to train and evaluate models under clean, static conditions, failing to assess how performance degrades when exposed to realistic perturbations [7,46,47]. This oversight creates a critical disconnect between academic results and practical deployment. Without robust evaluation, high accuracy on test datasets offers little assurance that a model will behave reliably under real-world uncertainty.

## 3. Methods

Figure 2 presents an overview of the proposed methodology, outlining the major steps from data preparation to robustness evaluation. The main steps include data preprocessing, input time-step design, perturbation generation and metric evaluation, followed by perturbation-based analysis of model sensitivity and robustness. Each component is described in detail in the following sections.

### 3.1. Data Description and Preprocessing

The dataset employed in this study was derived from the Long-Term Pavement Performance (LTPP) database, a nationally curated archive of field-tested pavement data. It integrates key variables collected from Falling Weight Deflectometer (FWD) measurements, ambient and subsurface climate sensors, and structural layer records, offering a comprehensive representation of in-situ pavement behavior under dynamic loading conditions.

The selected input features fall into four functional categories:Load and Environment: including the applied load magnitude (DROP_LOAD), pavement surface temperature (PVMT_SURF_TEMP), and ambient air temperature at the time of testing (AIR_TEMP_TEST).Deflection Responses: peak surface deflections at seven radial distances from the load plate (PEAK_DEFL_1 to PEAK_DEFL_7), representing structural stiffness variations with depth.Thermal Profile: subsurface layer temperatures at multiple depths (LAYER_TEMPERATURE_1 to LAYER_TEMPERATURE_3), reflecting thermal gradients affecting modulus.Structural Configuration: layer-specific thicknesses (BC_LAYER_THICKNESS_1 to _4) and material types (BC_LAYER_TYPE_1 to _4), indicating design heterogeneity across pavement cross-sections.

The predictive targets are four back-calculated modulus values, MODULUS_1 to MODULUS_4 (noted as M1 to M4 in the following context), representing the stiffness of the subgrade and overlying structural layers. These targets serve as proxies for mechanistic response indicators and are essential for data-driven evaluation of pavement conditions. To ensure analytical validity, the dataset was cleaned by removing records with missing entries or back-calculation errors with excessive residuals, as measured by root mean square error (RMSE) thresholds in the FWD dataset provided by the LTPP database. Only records with an ERROR_STATUS_EXP flag indicating “acceptable results from the backcalculation process” were retained, meaning that the RMSE value was acceptable and the elastic moduli were within the range typically observed for the specific material. This preprocessing established a controlled baseline for meaningful assessment of model performance [17,48] and provided a consistent starting point for evaluating model robustness through systematically applied perturbations.

Table 1 presents an example of a raw FWD test entry extracted from the LTPP database. The record includes the SHRP section identifier, test date, load level, deflections at seven geophone positions, back calculated modulus values, layer temperatures, layer thicknesses, and corresponding material type codes. This detailed composition highlights the nature of the dataset, which combines mechanical responses, thermal conditions, and structural configurations for each pavement section tested.

The final dataset used in this study contains 9000 valid FWD test records after preprocessing. These samples have been sourced from multiple U.S. states, with the majority originating from Alabama, Arizona, and Florida, followed by Colorado, Montana, and California. This geographic spread covers distinct climatic zones. In this paper, the effects of climatic zones on the FWD results are analyzed primarily using the measured temperature records from the raw FWD data while the effects of precipitation and wind are not included in the proposed models. The selected data records cover a wide range of temperatures, reflecting testing conducted in both winter and summer seasons and capturing thermal effects from low-temperature stiffening to high-temperature softening of asphalt layers. Testing conditions include measured pavement surface temperatures ranging from 3.0 °C and 51.6 °C and corresponding air temperatures between 3.0 °C and 37.7 °C. This selection ensures that the proposed model can operate effectively under seasonal variations with a broad range of temperature conditions. All samples in this dataset correspond to flexible pavements, as indicated by the surface layer material code (BC_LAYER_TYPE_1 = 1) for 100% of records. FWD test is most used to back-calculate layer moduli in flexible pavements. This approach ensures that the selected dataset reflects diverse geographic and environmental conditions relevant to flexible pavement performance, thereby supporting the generalizability of the proposed model across varied field conditions.

### 3.2. Time-Step Construction Strategies

While recurrent neural networks (RNNs) are well-known for their ability to model temporally ordered data, a unique challenge arises in this paper: the input features derived from Falling Weight Deflectometer (FWD) tests, although collected following a fixed protocol, do not exhibit an intrinsic temporal structure. Instead, they represent a combination of spatial, structural, and environmental characteristics. This raises a critical question: how should such features be organized to enable RNNs to capture meaningful dependencies?

To address this, we adopt an exploratory time-step construction approach, treating the input structuring process itself as an experimental design variable. Specifically, we evaluate five distinct sequence plans, namely Plan A to Plan E, each representing a different inductive hypothesis regarding the potential benefits of sequential ordering for model learning. Rather than relying on arbitrary flattening or expert-driven heuristics, we systematically examine which grouping schemes yield the best balance of predictive accuracy and robustness [49,50,51].

The five time-step construction strategies (Plan A to Plan E) were developed to explore how different representations of pavement-related inputs influence the model’s ability to learn temporal and structural dependencies. Each plan is guided by specific hypotheses grounded in the mechanical behavior of pavement systems under dynamic loading:Plan A simulates the vertical layering of pavement by sequentially grouping deflection and material characteristics by depth. This ordering reflects how stress waves propagate downward through the layers after impact, enabling the model to process information in the order that mechanical responses occur.Plan B organizes inputs by their physical categories, such as load, temperature, deflection, and structure, based on the assumption that temporally grouping homogeneous variables may enhance signal consistency and reduce feature interference across time steps.Plan C takes a region-based perspective, grouping load and near-load deflection readings together, followed by temperature and structure, and then far-field deflection inputs. This reflects a localized integration strategy that associates the load with the most affected pavement areas.Plan D is designed to follow the natural cause-and-effect chain in pavement behavior. It begins with external excitations (load and climate), progresses through intermediate mediators (thermal gradients), and then observes system responses (deflections), concluding with structural indicators (layer type and thickness). This sequencing allows the model to build temporal associations that mirror how mechanical processes unfold.Plan E modifies Plan A by combining far-field deflections into one time step, allowing assessment of how spatial distance in the deflection field affects model learning and robustness. This plan tests whether separating distant sensors improves temporal sensitivity or causes instability in the sequence.

These plans are not purely heuristic but reflect competing assumptions about how best to encode spatial, modal, and causal relationships into the input sequence. By systematically comparing their effects, this paper evaluates not just model accuracy, but also the alignment between sequence logic and real-world pavement behavior underload. Each constructed sequence was normalized using StandardScaler, reshaped into 3D tensors, and passed into the model architecture described in Section 3.3. Table 2 provides a comprehensive summary of how input features are distributed across time steps in each plan. By conducting this structural comparison within a controlled experimental protocol, we aim to establish whether temporal input design can compensate for the absence of explicit time dynamics in FWD data and enhance the practical applicability of sequence-based models in pavement analysis.

### 3.3. Deep Model Architecture

To systematically investigate how architectural design interacts with structured sequential input, we construct a hierarchical family of learning models with increasing complexity. This progression—from classical regressors to deep recurrent hybrids—aims to disentangle the specific contributions of model capacity, memory mechanisms, and feature fusion strategies to prediction accuracy and robustness under both clean and perturbed conditions [17,49]. Our modeling framework begins with traditional machine learning algorithms including Linear Regression, Bayesian ridge regression, support vector regression, k-nearest neighbors, random forest, gradient boosting, and XGBoost. These models operate on flattened input vectors and serve as interpretable and efficient baselines, albeit without support for temporal or spatial ordering in the data. To enhance representational power, we incorporate a fully connected Artificial Neural Network (ANN) that treats the input as a flat vector but captures nonlinear interactions through deep feedback. This model serves as a transitional baseline between traditional regressors and time-aware networks.

Building on these foundations, we design a series of recurrent neural architectures that leverage the structured time-step inputs proposed in this study. Starting with a SimpleRNN model that captures short-term dependencies, we progressively integrate more advanced components. A Wide & Deep (W&D) configuration combines temporal and static feature branches, enabling parallel learning of sequential patterns and contextual properties. Further, the sequential branch is enhanced by replacing SimpleRNN with a Long Short-Term Memory (LSTM) unit to improve long-range dependency modeling, or alternatively, with a Gated Recurrent Unit (GRU) for parameter efficiency. The most expressive architecture stacks LSTM and GRU layers within the deep branch to encode multi-scale memory dynamics while maintaining the W&D dual-path structure. All deep models are trained in a multi-output regression setting, with input sequences aligned according to domain-informed time-step plans. This design enables us to assess not only the raw accuracy of each model but also its stability and generalization behavior under synthetic perturbations that simulate real-world sensor noise [5,52,53,54,55].

#### 3.3.1. Classic Machine Learning Models

To establish non-sequential baselines, we implemented seven conventional regression models that process flattened input features without temporal structuring. These models are widely used in tabular prediction tasks and offer valuable benchmarks in terms of simplicity, interpretability, and efficiency. Although these models do not capture temporal dependencies, they are useful for quantifying the benefits of sequence-aware architectures under both clean and perturbed conditions [5,52,53,54]. Table 3 summarizes their key characteristics.

#### 3.3.2. Deep Learning Models

To explore the advantages of sequential representation and memory modeling, we implemented a series of deep neural networks with increasing structural complexity. All models were trained in a multi-output regression format to simultaneously predict MODULUS_1 to MODULUS_4. Table 4 summarizes the deep learning architecture used in this study. All RNN-based models received structured input sequences based on Plans A–E (as described in Section 3.2), standardized using StandardScaler and reshaped into 3D tensors. Training used mean-squared-error loss with early stopping and dropout regularization [55].

### 3.4. Perturbation Evaluation

To evaluate the robustness and deployment readiness of the proposed models under real-world uncertainties, we conducted a perturbation-based stability analysis by introducing synthetic noise into selected input features. This approach simulates typical disturbances such as sensor drift, calibration errors, and ambient variability that may affect FWD measurements in practical field settings. Specifically, we applied Gaussian noise with a standard deviation of ±3% to surface deflection measurements. This level was selected based on empirical observations by Rocha et al. (2004) [56], which reported that deflection variability typically ranges from below 2% to over 10%, depending on pavement conditions. The ±3% setting thus provides a conservative and statistically supported approximation of real-world noise. Gaussian noise was selected as a first-order approximation of sensor uncertainty due to its mathematical simplicity, its wide acceptance in engineering applications [57], and the lack of detailed noise profiles in the LTPP FWD datasets used in this study. While more complex noise structures could be explored in future work, Gaussian perturbation provides a generalizable and interpretable benchmark for evaluating model robustness. For each sample $x_{i}$, its perturbed version ${\bar{x}}_{i}$ is generated thus:(1)$${\bar{x}}_{i}=x_{i}\times \left(1+\epsilon_{i}\right)\;\;,\;\epsilon_{i}\sim N\left(0,0.003^{2}\right)$$

For instance, for a single FWD measurement, which consists of seven deflections recorded by geophones, Gaussian noise with a ±3% standard deviation is added to each of the seven deflections. This process is repeated 10 times for each FWD measurement, generating 10 noisy variants per instance. The perturbed samples are then fed into the trained model to obtain a distribution of predictions.

#### Evaluation Metrics

To quantify the model’s response to noise and evaluate its predictive consistency, we compute the following statistical indicators for each selected model.

(1)Mean Prediction (μ)

The mean of predicted values reflects the model’s average response under noise perturbation:(2)$$\mu \;=\frac{1}{N}\sum_{i\;=1}^{n}{\hat{y}}_{i}$$

Here, ${\hat{y}}_{i}$ is the predicted value at trial i and N = 10 is the number of predictions per sample. A mean prediction value closer to the ground truth indicates higher overall accuracy. This metric itself does not measure variability but provides the baseline for interpreting other robust metrics.

(2)Standard Deviation (σ)

The standard deviation represents the prediction dispersion, which captures prediction uncertainty under noise:(3)$$\sigma =\sqrt{\frac{1}{N}\sum_{i\;=1}^{N}{\left({\hat{y}}_{i}-\mu \right)}^{2}}$$

Smaller σ values indicate that the model’s predictions are more consistent across noise-perturbed inputs, which implies higher robustness.

(3)95% Confidence Interval (CI) Width

Assuming normality, the 95% confidence interval width is approximated thus:(4)$$CI_{95\%}\;=\;2\times 1.96\times \sigma$$

This metric quantifies the range within which predictions are expected to fall with 95% probability. Narrower CI widths indicate greater stability and less uncertainty in model predictions.

(4)Coverage Ratio

This metric evaluates the proportion of ground truth values that fall within their respective confidence intervals:(5)$$Coverage\;=\;\frac{1}{M}\sum_{j\;=\;1}^{M}I\left(y_{j}\in \left[\mu_{j}-1.96\sigma_{j},\mu_{j}+1.96\sigma_{j}\right]\right)$$

Here, $y_{j}$ is the ground truth, *M* is the number of test samples, and I (⋅) is the indicator function. Higher coverage values (closer to 1) indicate that the model’s uncertainty estimates align well with actual outcomes, which is desirable for deployment in noisy environments.

## 4. Experiments and Results

The previous sections have established the theoretical foundations, architectural designs, and methodological innovations that underpinned this study. In this section, we present a comprehensive set of experiments that quantified how input structuring, model architecture, and data perturbation jointly affect the accuracy and robustness of pavement modulus prediction. This section addresses key research questions introduced earlier: How does time-step structuring influence sequence model performance? Can physically informed input design mitigate the impact of noise? Which architectures best balance predictive precision with generalizability? By integrating perturbation-based testing and temporal input variation within a unified experimental framework, this paper not only benchmarks model performance under ideal conditions but also reveals how models behave under uncertainty.

### 4.1. Model Configurations and Experimental Design

This paper assesses how temporal input structuring and architectural complexity influence pavement modulus prediction through a comprehensive experimental framework that integrates multiple models, input strategies, and evaluation criteria. The goal is to evaluate model accuracy, generalization, and robustness under the perturbed environments of deployable predictive systems [58]. Before adopting a unified temporal structuring scheme for deep learning experiments, we compared five time-step input strategies (Plan A through Plan E) as outlined in Section 3.2. The ResRNN-Transformer architecture, with consistent training conditions and data (3000 samples), was used to evaluate the predictive performance of each plan across four target moduli (M1–4). The results showed that Plan D consistently outperformed others in both coefficient of determination (*R*^2^) and mean absolute error (MAE) across three out of four targets. Specifically, for M1 and M2, Plan D achieved *R*^2^ values of 0.75 and 0.76, with MAE of 0.34 and 0.33, respectively, outperforming the next best plan (Plan B) whose *R*^2^ values for M1 and M2 were 0.58 and 0.74. For M3, Plan D also recorded a higher *R*^2^ (0.68) compared to Plan B (0.64) while maintaining a lower MAE (0.43 vs. 0.46). This top–down arrangement—beginning with the environmental context and progressing toward structural and deflection features—appeared to align more effectively with the causal structure of pavement response mechanisms. Detailed results are provided in Table 5.

The effectiveness of Plan D stems from its close alignment with the actual causal sequence present in pavement behavior under FWD loading. This sequencing begins with global contextual inputs such as the applied load and ambient temperature, followed by subsurface temperature gradients, then transitions into deflection responses near the loading point, and concludes with structural layer properties. This arrangement reflects the physical process of stress wave propagation through pavement layers, ensuring that the model receives foundational stimuli before processing the resulting mechanical reactions. Additionally, the structural layer properties represent inherent pavement characteristics that remain unaffected by FWD testing. By preserving this causality, the time-series model can develop a more coherent internal representation of input–output relationships. Conversely, alternative input orders that blend or rearrange these categories tend to obscure physical meaning and reduce learning consistency. In experimental trials, Plan D consistently led to faster convergence and narrower prediction uncertainty, particularly when inputs were disturbed, suggesting that its physically grounded ordering enhances both stability and interpretability.

Therefore, Plan D was selected as the default input sequence design for subsequent experiments involving temporal models. All experiments followed a unified data preprocessing pipeline, which included missing value imputation, categorical encoding, and numerical normalization. Two input configurations were compared: (1) Deflection-only features and (2) All-Features, which included load magnitude, thermal gradients, structural layer composition, and deflections. For sequence-based models, the Plan D structure was employed to transform tabular features into six-step temporal sequences, capturing a top–down physical hierarchy from environmental inputs to deep structural responses. In contrast, traditional machine learning models and the feedforward ANN consumed flattened vectors.

Three dataset scales, 3000, 6000, and 9000 samples, were employed to assess scalability and learning behavior under different data availability scenarios. Given the sparse and site-dependent nature of test data in geotechnical engineering, understanding how model performance varies with dataset size is crucial. All deep learning models were implemented using TensorFlow/Keras APIs and trained in a reproducible CPU-only environment (Google Colab with high-RAM runtime, backed by a Lenovo Legion Y9000 laptop, Python 3.10). Each model was trained for 200 epochs using the Adam optimizer (learning rate = 0.001), with early stopping and dropout applied to mitigate overfitting. A 70/30 train-test split was used throughout, and random seed 42 was set for full reproducibility.

Among the evaluated architectures, the ResRNN-W&D + LSTM + GRU model exhibited the longest training time, reaching 235.54 s on the largest dataset (9000 samples), due to its increased parameter depth and stacked recurrent layers. While hybrid models offer improved robustness and accuracy, they inevitably introduce greater computational overhead compared to simpler architectures. The complete implementation details and computational comparisons for all models, including training time statistics across different data scales, were documented in a related study [59], which served as a technical supplement to the present work [59].

#### 4.1.1. Traditional Machine Learning Models

Traditional machine learning models (e.g., SVM, random forests) typically operate on static, vectorized input representations and do not model temporal dependencies, unless explicit feature engineering is applied. While inherently limited in capturing sequential dynamics, these models offer strong advantages in interpretability, computational efficiency, and suitability for tabular regression tasks. As outlined in Table 3, seven such models were implemented in this study as baseline comparators. Their inclusion provides a critical reference point for evaluating the added value of temporal abstraction and architectural depth introduced by sequential deep learning approaches.

#### 4.1.2. Deep Feedforward Neural Model

A fully connected Artificial Neural Network (ANN), as detailed in Table 4, was implemented as a deep learning baseline. It processes flattened input features in a multi-output regression setup, offering a transitional bridge between shallow ML models and sequential RNN architectures.

#### 4.1.3. Deep Sequential and Hybrid Architectures

To capture temporal dependencies and hierarchical structure within pavement response data, we implemented a family of deep learning architectures that leverage time-step organized inputs based on Plan D. These include recurrent models such as SimpleRNN, hybrid Wide & Deep configurations, and advanced stacked variants incorporating GRU, LSTM, and residual connections, as summarized in Table 4. Each model was trained and evaluated independently across three dataset sizes (3000, 6000, and 9000), with 10 repeated runs per configuration to ensure statistical reliability. Performance was assessed using four standard metrics—MAE, MSE, *R*^2^, and runtime—aggregated to evaluate both predictive accuracy and consistency. This unified evaluation framework enabled a systematic comparison across architectures under controlled conditions, providing insights into how model depth, memory mechanisms, and input fusion strategies influence robustness and generalization in noise-prone, real-world scenarios [60,61].

### 4.2. Model Comparison

Table 6 and Table 7 show the computed Mean Absolute Error (MAE), Mean Squared Error (MSE), and Coefficient of Determination (*R*^2^) values for four target outputs: MODULUS_1, MODULUS_2, MODULUS_3, and MODULUS_4 across both ML and DL models. The equations of *R*^2^, the MAE, and the MSE are shown in Equations (6) and (8). The performance of each ML and DL model is discussed below, based on varying data scales.

(1)Coefficient of Determination ($R^{2}$)

This metric evaluates how well the predicted values approximate the ground truth. It is calculated thus:(6)R2 = 1−Σi = 1Myi−y^i2Σⅈ = 1Myi−y¯2

Here, $y_{i}$ is the ground truth, ${\hat{y}}_{i}$ is the predicted value, and $\bar{y}$ is the mean of the ground truth.

(2)Mean Absolute Error (MAE)

MAE measures the average magnitude of prediction errors:(7)MAE = 1MΣⅈ = 1My^i−yi

(3)Mean Squared Error (MSE)

MSE penalizes larger errors more than MAE, and is defined thus:(8)MSE = 1MΣⅈ = 1My^i−yi2

#### 4.2.1. Performance Comparison on 3000-Sample Dataset

In the limited data scenario, both traditional ML and DL models exhibited constrained generalization. However, distinct performance trends emerged that highlight the structural advantages of certain architectures. Among ML models, Random Forest, Gradient Boosting, and XGBoost consistently outperformed simpler baselines, achieving *R*^2^ values exceeding 0.75 for M1 and M2. For instance, XGBoost achieved *R*^2^ values of 0.80 for M2 and 0.85 for M4, with corresponding MAE values of 0.05 for both, outperforming Random Forest, which recorded *R*^2^ values of 0.78 (M1) and 0.80 (M2) with MAE of 0.06 and 0.05, respectively. Simpler models such as Linear Regression lagged significantly, with *R*^2^ as low as 0.35 for M3. However, all ML models struggled with M3, where *R*^2^ often fell below 0.70, suggesting difficulty in capturing complex dependencies. Notably, SVR’s performance declined as the dataset size increased. While it achieved *R*^2^ = 0.78 on the 3000-sample dataset, performance dropped to −3.96 with 6000 samples and −1.7 with 9000 samples. This suggests SVR struggled to generalize with increasing data complexity, likely due to kernel limitations and sensitivity to high-dimensional features. Despite data filtering, strong multicollinearity among deflection measurements (e.g., PEAK_DEFL_1–4) may have hindered SVR ‘s learning process. This paper focuses on comparing performance trends across ML and DL models under varying data availability rather than optimizing individual model accuracy. To ensure consistent and interpretable comparisons, all ML models were evaluated using their default or minimally adjusted settings, reflecting common engineering practices where model selection prioritizes robustness and ease of use, particularly during early deployment stages. Consequently, models like SVR were not extensively tuned, and their results reflect their default behavior across different data sizes.

In contrast, DL models, especially those with hybrid architecture, demonstrated superior performance. The ResRNN-Wide&Deep + LSTM + GRU model achieved the best overall results, with *R*^2^ values of 0.82, 0.83, 0.76, and 0.93 for M1 through M4, and corresponding MAE values of 0.27, 0.27, 0.33, and 0.17, respectively. Compared to the base SimpleRNN model (*R*^2^ of 0.35, 0.40, 0.34, and 0.61), the hybrid design reduced the MAE by more than 50% across all targets. Compared to the base SimpleRNN model—which severely underfitted M3 (*R*^2^ < 0.35)—the addition of wider paths and deeper recurrent units (i.e., GRU, LSTM) substantially enhanced the model’s ability to extract temporal and structural features, even from short sequences.

These findings underscore a key insight: deep hybrid models are better suited to learn temporal dependencies and interlayer physical patterns under small-sample conditions. While traditional ML models remain competitive for more straightforward outputs, they inherently lack the representational capacity to fully capture the sequential and hierarchical nature of pavement modulus data. Moving forward, the integration of temporal modeling with architectural depth emerges as a crucial direction for robust prediction in data-limited civil infrastructure scenarios.

#### 4.2.2. Performance Comparison on 6000-Sample Dataset

With a larger dataset, model performance improved significantly, allowing clearer differentiation between models. Among traditional ML models, Random Forest, Gradient Boosting, and XGBoost continued to perform well, with *R*^2^ ≥ 0.98 across most targets. These ensemble models demonstrated strong generalization capacity when trained on well-structured, mid-sized datasets. However, simpler models like Linear Regression and SVR remained more sensitive to nonlinearity, with *R*^2^ values generally lower and more variable across outputs.

In deep learning, performance gains were even more significant. The ResRNN-Wide&Deep + LSTM + GRU model reached *R*^2^ > 0.99 for M1 through M4, delivering near-perfect predictions and outperforming all other models. GRU-based architectures, such as ResRNN-W&D + GRU, also performed robustly with *R*^2^ ≥ 0.97, demonstrating the effectiveness of simplified gated units. Even the base ResRNN-W&D structure surpassed most ML models, consistently achieving *R*^2^ above 0.90 across all modulus layers. Both ML and DL models performed better with increased data availability, but deep hybrid models gained a stronger edge by leveraging temporal abstraction. While ML models efficiently captured shallow-to-mid-level patterns, deep models began to fully utilize sequence encoding mechanisms to extract cross-layer physical interactions, particularly in more complex targets like M3 and M4.

Under mid-scale data conditions, ensemble ML models remain competitive and efficient, but deep hybrid architectures clearly surpass them in capturing long-range dependencies and structural intricacies. This stage marks a transition where data sufficiency enables temporal models to express their full potential, laying the groundwork for robust deployment in infrastructure prediction tasks.

#### 4.2.3. Performance Comparison on 9000-Sample Dataset

With the largest dataset, both ML and DL models approached peak performance, with accuracy stabilizing across all four modulus targets. The ResRNN-W&D + LSTM + GRU + Transformer architecture demonstrated the best overall performance, reaching *R*^2^ > 0.99 for all outputs and near-zero MAE and MSE values, showcasing its exceptional learning capacity in high-data regimes. Ensemble ML models like XGBoost and Random Forest performed comparably well, achieving *R*^2^ ≈ 1.00 for most outputs, particularly for well-behaved targets like M1 and M2.

Despite improvements across all models, the SimpleRNN model remained the weakest performer, particularly for complex outputs like M3 and M4, where *R*^2^ stayed below 0.90. This highlights the limitations of shallow recurrence without gating or attention mechanisms for modeling complex structural dependencies. The findings confirm that model scalability is closely linked to data richness. While both advanced deep architectures and tuned ensemble ML models achieve near-perfect predictions in large-sample scenarios, deep hybrid models maintain an advantage in capturing temporal abstractions and latent nonlinearities, which are crucial for modeling the dynamic behavior of real-world pavements. These results show a clear trend: model performance improves consistently with increased structural depth. The use of stacked temporal encoding, residual integration, and multi-head attention enhances both robustness and precision, particularly as the data volume grows.

### 4.3. Comparative Analysis of Model Robustness and Input Feature Structure

To assess model robustness under data perturbation and examine the impact of input feature structure (Deflection-only vs. All-Features) on predictive performance, this section evaluates both ML and DL models under ±3% Gaussian noise. The models tested include (1) traditional machine learning models—Gradient Boosting and XGBoost and (2) deep neural network models—ResRNN-W&D + GRU, RNN-W&D + GRU + LSTM, and ResRNN-W&D + GRU + LSTM as these models demonstrated better predictive performance based on the results presented in Table 6 and Table 7.

Gaussian noise was introduced to the seven peak surface deflection measurements (PEAK_DEFL_1 to PEAK_DEFL_7) to simulate sensor drift and environmental fluctuations. As shown in Figure 3, the perturbed deflections (red) exhibited small but consistent deviations from the original measurement (green), forming the basis for robust analysis across models and input configurations.

Each model was tested under two separate input configurations: Deflection-only and All-Features. A comparative analysis was performed, based on prediction distributions, accuracy metrics, and sensitivity to perturbation, to evaluate model stability and generalization. Since Gaussian noise was applied directly to the deflection measurements, evaluating models with the Deflection-only features serves two purposes: first, it highlights the direct impact of perturbation on the most noise-exposed variables, and second, it reflects the common approach of using only deflection data from FWD tests in many previous studies without incorporating auxiliary information. Comparing these results against the All-Features configuration allows for an empirical assessment of whether including additional thermal, structural, and contextual variables improve robustness under input disturbance [62,63].

#### 4.3.1. Overall Assessment: Deflection-Only vs. All-Features Input Structures

Table 8 presents the average original pavement moduli from the dataset used in the perturbation analysis. These values serve as the benchmark for evaluating the performance of both ML and DL under ±3% Gaussian noise. Since 50 records were used as the baseline for generating perturbed inputs, individual modulus values for each record are not listed in order to avoid repetition and improve readability.

Table 9 summarizes the predicted mean, standard deviation, confidence interval (CI) width, and coverage ratio, calculated using the equations outlined in Section Evaluation Metrics, for four pavement layers (MODULUS_1 to MODULUS_4) using Gradient Boosting and XGBoost models under two input scenarios: Deflection-only and Plan D. Under the Deflection-only configuration, XGBoost produced substantially wider confidence intervals—for example, 1409.04 for M1 and 837.88 for M2—but achieved high coverage across all moduli (86% for both M1 and M2, and above 82% for M3 and M4). This indicates a more conservative and reliable uncertainty estimation. In contrast, Gradient Boosting yielded narrower intervals but exhibited lower coverage ratios (16–30%), which suggests a tendency to underestimate prediction uncertainty.

When using Plan D inputs, both models showed reduced variability. For example, in XGBoost, the CI width for M1 decreased from 1409.04 (Deflection-only) to 572.88 while coverage dropped from 86% to 70%. Similarly, Gradient Boosting’s CI width for M2 shrank from 278.37 to 66.10, but coverage fell from 28% to just 2%. Gradient Boosting produced extremely narrow intervals accompanied by very low coverage ratios, as low as 2% in some cases, highlighting severe underconfidence. XGBoost, on the other hand, maintained relatively high coverage, between 70 and 80%, while reducing interval widths. This indicates a more balanced performance in terms of robustness and reliability.

Overall, XGBoost provided better-calibrated prediction intervals across both input configurations, particularly for the deeper layers MODULUS_3 and MODULUS_4. While the Plan D input configuration yielded narrower confidence intervals for XGBoost, it also slightly decreased the coverage. This suggests a trade-off: Plan D offers narrower predictions whereas Deflection-only inputs provide more conservative and reliable uncertainty bounds. The choice between input configurations should therefore be informed by the specific requirements of the engineering application, depending on whether it prioritizes prediction accuracy or uncertainty robustness. From an engineering perspective, this trade-off has practical implications. The Plan D configuration, by producing tighter prediction bounds, may support more cost-efficient pavement designs when confidence in input data is high. However, the Deflection-only configuration, with its higher coverage ratios, may be more appropriate in high-risk conditions where uncertainty quantification is critical. These conditions may include high-traffic routes, areas with poor subgrade conditions, or early-stage designs with limited field data. Therefore, input structure selection should be guided not only by model performance metrics but also by the characteristics of the input uncertainty and the acceptable level of risk in the engineering context.

Table 10 presents the prediction performance of deep learning models using both Deflection-only and Plan D inputs. These results can be directly compared with the traditional machine learning models summarized in Table 9. In general, deep learning models demonstrated higher coverage ratios than Gradient Boosting when Deflection-only inputs were used. With Plan D inputs, deep learning models consistently achieved coverage ratios above 0.65 across the pavement layers, outperforming both Gradient Boosting and XGBoost in most cases. However, the improved coverage of deep learning models was accompanied by wider confidence intervals and greater prediction variability, particularly for MODULUS_1 and MODULUS_2. Traditional models such as XGBoost produced narrower uncertainty bounds under the Deflection-only configuration, often with comparable or even higher coverage ratios. For example, XGBoost achieved a confidence interval width of 1409.04 at a coverage ratio of 86% while ResRNN-W&D + GRU + LSTM had a wider interval of 1751 with a slightly lower coverage ratio of 84%.

The results of this perturbation-based analysis indicate that XGBoost performs more effectively with Deflection-only inputs whereas deep learning models exhibit superior performance when Plan D inputs are utilized. This distinction may be attributed to the lower dimensionality and clearer structure of the deflection data, which align well with the capabilities of traditional ML models. In small datasets, these models tend to generalize better, offering more stable and narrower prediction intervals. In contrast, deep learning models may struggle to extract meaningful patterns from limited, low-dimensional input, especially when those inputs are directly perturbed. As a result, they exhibit wider confidence intervals and less stable performance in this configuration.

#### 4.3.2. Model Sensitivity to Subgrade Strength

The models analyzed in Section 4.3.1 were originally trained using data from strong subgrade conditions. To assess their generalizability and stability under more challenging environments, this section applies those models to weak subgrade conditions. By comparing performance across weak and strong subgrade scenarios, we can obtain a deeper understanding of model robustness under varying ground conditions. Subgrade stiffness is a critical factor in determining pavement performance, and weak subgrades, which are typically characterized by low bearing capacity, are frequently encountered in pavement engineering. If not properly accounted for during design or maintenance, such subgrades can accelerate rutting, induce structural fatigue, and lead to premature pavement failure. Evaluating predictive performance under different subgrade strengths provides valuable insight into model reliability and generalizability, ultimately supporting the development of more resilient and cost-effective pavement systems. Table 11 presents the average original pavement moduli from the dataset with weak subgrade conditions used in the perturbation analysis. Again, a total of 50 records served as the baseline for generating perturbed inputs. Individual modulus values for each record are not shown to avoid redundancy and to improve readability. Table 12 and Table 13 present the prediction from these models on weak subgrade conditions. Across both traditional ML and DL models, predictions for strong subgrades generally exhibited lower variability and tighter confidence intervals than those for weak subgrades. This aligns with the expected behavior in pavement engineering, where stiffer foundations tend to produce more stable and predictable structural responses.

Among the traditional models, Gradient Boosting was the most sensitive to subgrade condition changes. Under weak subgrade scenarios, its coverage ratios dropped from 30% (M1, strong subgrade) to 27%, and from 28% (M2, strong) to 29%, with CI widths remaining narrow at around 612 for M1, indicating a strong underestimation of uncertainty. In contrast, XGBoost maintained high coverage—for example, 98% for M1 and 84% for M2 in weak subgrades—despite slightly wider CIs (1376.71 for M1, 785.83 for M2). Under strong subgrade conditions, XGBoost’s coverage remained consistently high (≥84% across M1–M4), whereas Gradient Boosting showed only moderate improvement compared to the weak subgrade case. This indicates that Gradient Boosting is less robust under data perturbation, particularly when the subgrade conditions are less favorable. In comparison, XGBoost demonstrated much better robustness, maintaining relatively high coverage ratios (0.70–0.90) across both subgrade conditions. While its confidence intervals were wider under weak subgrade scenarios, the prediction ranges remained reasonable and consistent, indicating that XGBoost provides a more balanced trade-off between accuracy and uncertainty. Importantly, its performance under weak subgrades was comparable to that of deep learning models, such as ResRNN-W&D + GRU + LSTM, suggesting that XGBoost can generalize well even when applied to more challenging, low-stiffness environments. Deep learning models also showed strong generalizability across subgrade conditions. Although their confidence intervals widened under weak subgrade inputs, the coverage ratios remained stable and relatively high, often exceeding 0.70. This indicates that deep learning models provide more calibrated uncertainty estimates and can maintain reliable predictions in the face of increased ground variability.

In summary, while both XGBoost and deep learning models performed consistently across subgrade types, Gradient Boosting exhibited the greatest performance degradation under weak subgrade conditions. These findings underscore the importance of robust uncertainty modelling in soft soil scenarios and suggest that models like XGBoost and RNN-based deep learning offer better reliability in diverse geotechnical environments.

#### 4.3.3. Key Takeaways from Perturbation-Based Evaluation

This subsection summarizes the key observations derived from the perturbation experiments conducted in Section 4.3. The goal is to distill generalizable patterns related to model stability, sensitivity, and input structure effectiveness under synthetic noise conditions.

(1)Deep hybrid models exhibit strong and consistent robustness. Across all four modulus outputs, architectures that integrate recurrent memory units (e.g., GRU, LSTM) with wide-path static inputs consistently produced the most stable predictions under ±3% Gaussian noise. Among the deep learning models, the ResRNN-W&D + GRU + LSTM architecture achieved the narrowest prediction intervals and demonstrated the least sensitivity to input perturbations.(2)Deflection-only inputs enhance robustness for ML models. Despite their lower dimensionality, deflection-only configurations often resulted in higher coverage ratios. This can be attributed to the ability of traditional ML models to generalize more effectively in low-dimensional, small-sample settings and to maintain stable performance when perturbations are directly applied to the input features.(3)The model architecture contributes more to robustness than input dimensionality. Advanced deep learning models maintained consistent performance across both Deflection-only and All-Features configurations. This suggests that architectural design, particularly the incorporation of memory units and the combination of wide and deep input pathways, plays a more critical role in enhancing model robustness than simply increasing the number of input features.

## 5. Conclusions and Future Work

This study systematically examined the robustness of various predictive models, including traditional machine learning approaches and several deep neural network architectures, under data perturbation scenarios for pavement modulus prediction. By developing a unified experimental framework and introducing ±3% Gaussian noise to simulate data uncertainty, we proposed a comprehensive robustness evaluation strategy that integrates both structural and temporal input variations. This approach offers novel insights for model development and deployment in pavement engineering applications.

The results highlight that input sequence design plays a critical role in determining the performance and robustness of time series models. Traditional evaluation metrics such as *R*^2^, the MAE, and the MSE were used to assess model accuracy while average prediction fluctuation, confidence interval width, and variance were employed to quantify robustness under input perturbation. Among the five input configurations (Plan A to Plan E), Plan D, which incorporates global contextual information and hierarchical temporal features, consistently produced the lowest prediction errors and the most stable performance. This was especially evident in RNN-W&D models incorporating GRU or LSTM layers. Deep learning models trained with the All-Feature set outperformed those using only deflection measurements. These findings are highly relevant to geotechnical modelling tasks, where measurement uncertainty and spatiotemporal variability present significant challenges to predictive accuracy and model reliability.

This work contributes to the growing body of research supporting smart infrastructure systems by enabling more reliable, adaptive, and uncertainty-aware prediction frameworks. In the context of smart cities, robust and interpretable pavement models are essential for real-time condition assessment, predictive maintenance, and data-driven asset management. The proposed perturbation-based methodology can be extended into a standardized robustness testing platform applicable to broader civil engineering domains, including geotechnical risk assessment and infrastructure health monitoring. Future research should incorporate real-world sensor noise and heterogeneous multi-source datasets to improve generalizability and explore uncertainty-aware learning architectures such as Bayesian neural networks to support smart and resilient infrastructure systems.

Nonetheless, several limitations of this study should be acknowledged. First, all experiments were conducted using data extracted from the LTPP database. To enhance generalizability, future research should examine datasets from other countries or pavement monitoring systems. Second, due to limitations in computational resources, model training and validation were carried out in a CPU-only environment. This setup may not have accurately represented the scalability or efficiency of the models under real-time or high-throughput deployment scenarios. Third, although the proposed perturbation framework effectively evaluated model robustness under controlled synthetic Gaussian noise, field validation using inputs affected by real world noise such as autocorrelated drift, systematic bias, or abrupt environmental disturbances was not included in this study due to data unavailability. Future work should include such validation when suitable datasets become accessible. These limitations highlight the importance of further investigation under a broader range of conditions to support practical implementation.

## Figures and Tables

**Figure 1 sensors-25-05222-f001:**
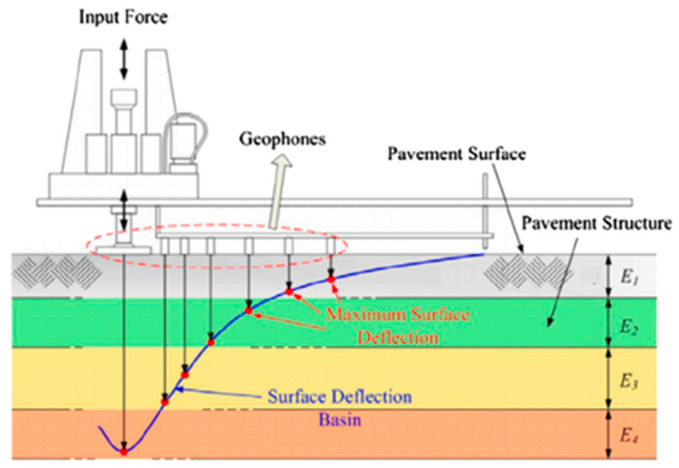
A schematic diagram of the FWD test [17].

**Figure 2 sensors-25-05222-f002:**
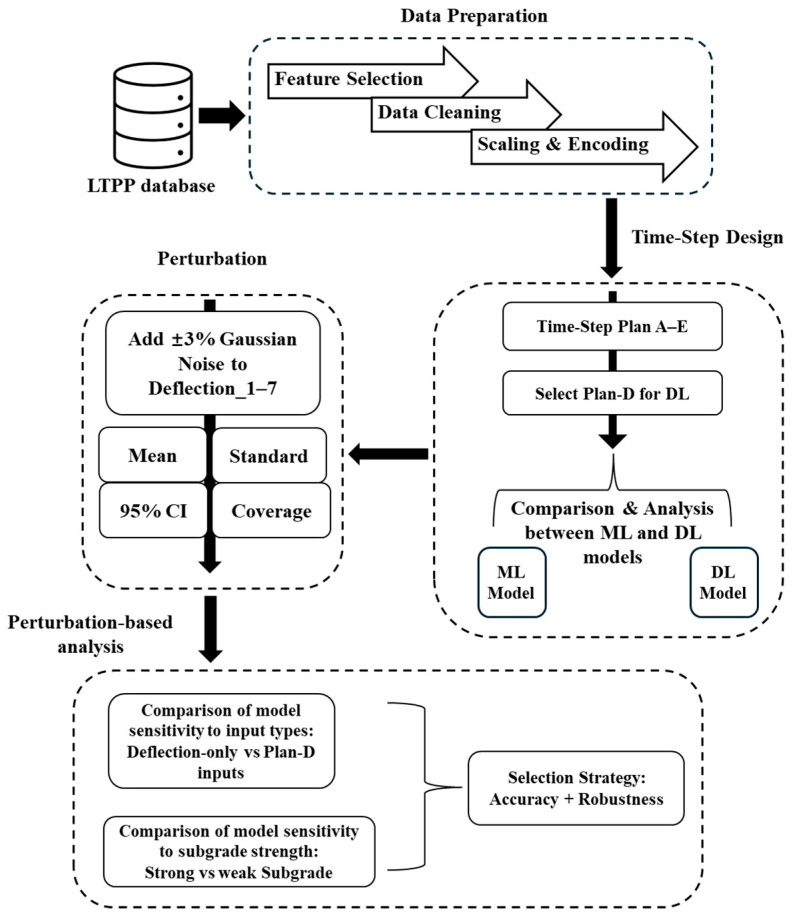
Overview of the proposed methodology.

**Figure 3 sensors-25-05222-f003:**
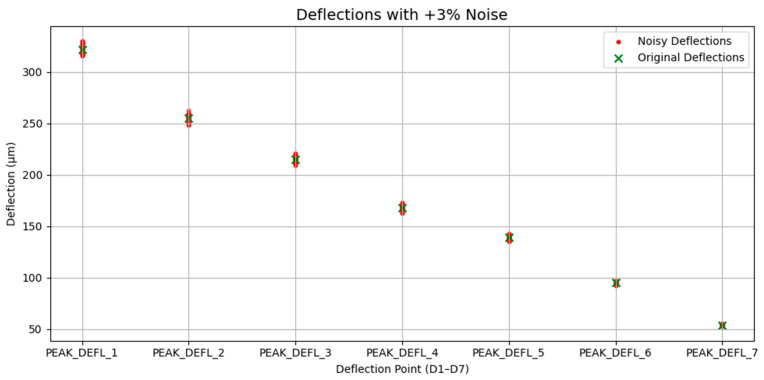
Visualization of deflection profile under Gaussian noise perturbation.

**Table 1 sensors-25-05222-t001:** Example of a raw FWD test record from the LTPP database.

SHRP_ID 12-508	Description					Time	
	Drop Height Position 2, Target Load 40 KN (9 Kips)—570 kPa for Standard LTPP Pavement Surface Tests					26 April 2004	
Actual drop pressure (kPa)	557						
DEFL (microns)	DEFL_1	DEFL_2	DEFL_3	DEFL_4	DEFL_5	DEFL_6	DEFL_7
	124	92	73	59	48	36	29
Modulus (ksi)	5037.5	437.8	650.9	261.3			
Temp (°C)	45.2	44.7	40.5				
Depth (mm)	247.3	157.5	609.6				
TYPE	1	9	23	23			

**Table 2 sensors-25-05222-t002:** Overview of time-step construction strategies (Plan A to Plan E).

Plan A			
Time Step	Step Description	Feature Count	Feature Fields
T0	Load and surface climate input	3	DROP_LOAD, PVMT_SURF_TEMP, AIR_TEMP_TEST
T1	Deflection and material info (Zone 1)	4	PEAK_DEFL_1, LAYER_TEMPERATURE_1, BC_LAYER_TYPE_1, BC_LAYER_THICKNESS_1
T2	Deflection and material info (Zone 2)	4	PEAK_DEFL_2, LAYER_TEMPERATURE_2, BC_LAYER_TYPE_2, BC_LAYER_THICKNESS_2
T3	Deflection and material info (Zone 3)	4	PEAK_DEFL_3, LAYER_TEMPERATURE_3, BC_LAYER_TYPE_3, BC_LAYER_THICKNESS_3
T4	Deflection and material info (Zone 4)	3	PEAK_DEFL_4, BC_LAYER_TYPE_4, BC_LAYER_THICKNESS_4
T5	Residual deflection signal	1	PEAK_DEFL_5
T6	Residual deflection signal	1	PEAK_DEFL_6
T7	Residual deflection signal	1	PEAK_DEFL_7
Plan B			
Time Step	Step Description	Feature Count	Feature Fields
T0	Load magnitude	1	DROP_LOAD
T1	Surface and air temperature	2	PVMT_SURF_TEMP, AIR_TEMP_TEST
T2	Subsurface temperature measurements	3	LAYER_TEMPERATURE_1, LAYER_TEMPERATURE_2, LAYER_TEMPERATURE_3
T3	Full deflection inputs	7	PEAK_DEFL_1~PEAK_DEFL_7
T4	Structural composition layer types	4	BC_LAYER_TYPE_1~BC_LAYER_TYPE_4
T5	Structural composition	4	BC_LAYER_THICKNESS_1~BC_LAYER_THICKNESS_4
Plan C			
Time Step	Step Description	Feature Count	Feature Fields
T0	Load and near-load deflection readings	3	DROP_LOAD, PEAK_DEFL_1, PEAK_DEFL_2
T1	Subsurface temperature profile	3	LAYER_TEMPERATURE_1, LAYER_TEMPERATURE_2, LAYER_TEMPERATURE_3
T2	Structural composition info	6	BC_LAYER_TYPE_1~BC_LAYER_TYPE_3, BC_LAYER_THICKNESS_1~BC_LAYER_THICKNESS_3
T3	Surface and ambient temperature	2	PVMT_SURF_TEMP, AIR_TEMP_TEST
T4	Aggregated deflection readings	5	PEAK_DEFL_3~PEAK_DEFL_7
Plan D			
Time Step	Step Description	Feature Count	Feature Fields
T0	Load magnitude	1	DROP_LOAD
T1	Surface and air temperature	2	PVMT_SURF_TEMP, AIR_TEMP_TEST
T2	Subsurface temperature profile	3	LAYER_TEMPERATURE_1, LAYER_TEMPERATURE_2, LAYER_TEMPERATURE_3
T3	Near-load deflection data	3	PEAK_DEFL_1, PEAK_DEFL_2, PEAK_DEFL_3
T4	Far-field deflection data	4	PEAK_DEFL_4~PEAK_DEFL_7
T5	Structural attributes	8	BC_LAYER_TYPE_1~BC_LAYER_TYPE_4, BC_LAYER_THICKNESS_1~BC_LAYER_THICKNESS_4
Plan E			
Time Step	Step Description	Feature Count	Feature Fields
T0	Load and climate input	1	DROP_LOAD, PVMT_SURF_TEMP, AIR_TEMP_TEST
T1	Local deflection and material info (Zone 1)	4	PEAK_DEFL_1, LAYER_TEMPERATURE_1, BC_LAYER_TYPE_1, BC_LAYER_THICKNESS_1
T2	Local deflection and material info (Zone 2)	4	PEAK_DEFL_2, LAYER_TEMPERATURE_2, BC_LAYER_TYPE_2, BC_LAYER_THICKNESS_2
T3	Local deflection and material info (Zone 3)	4	PEAK_DEFL_3, LAYER_TEMPERATURE_3, BC_LAYER_TYPE_3, BC_LAYER_THICKNESS_3
T4	Local deflection and material info (Zone 4)	3	PEAK_DEFL_4, BC_LAYER_TYPE_4, BC_LAYER_THICKNESS_4
T5	Consolidated residual deflections	3	PEAK_DEFL_5, PEAK_DEFL_6, PEAK_DEFL_7

**Table 3 sensors-25-05222-t003:** Overview of classic machine learning models.

Model Name	Description
Linear Regression (LR)	A standard parametric model that estimates outputs via weighted linear combinations of features.
Bayesian Ridge Regression	A probabilistic linear model that incorporates Bayesian regularization to improve stability under multicollinearity.
K-Nearest Neighbors (KNN)	A non-parametric model that makes predictions based on local averaging in feature space.
Support Vector Regression (SVR)	A kernel-based model that learns complex functional relationships within a maximum-margin framework.
Random Forest (RF)	An ensemble of decision trees using bootstrapping and averaging, with built-in feature importance metrics.
Gradient Boosting (GBR)	A sequential ensemble method that builds trees to minimize residual errors using additive modeling.
Extreme Gradient Boosting (XGBoost)	An optimized gradient boosting method with enhanced regularization, pruning, and parallel computation.

**Table 4 sensors-25-05222-t004:** Overview of deep learning models.

Model Name	Description
Artificial Neural Network (ANN)	A fully connected feedforward network that treats input as a flat vector. Serves as a baseline deep learning model without sequence awareness.
SimpleRNN	Introduces temporal learning by reshaping inputs into time-step sequences and applying a basic recurrent layer (SimpleRNN) to model short-term dependencies.
RNN + Wide & Deep (W&D)	Splits the input into two parallel streams: a “deep” branch for sequential data and a “wide” branch for static contextual features. Enables modeling of both time-sensitive and time-invariant information.
RNN-W&D + LSTM	Replaces the SimpleRNN unit with a Long Short-Term Memory (LSTM) layer to improve the capture of long-range dependencies across time steps.
RNN-W&D + GRU	Substitutes the LSTM with a Gated Recurrent Unit (GRU) to assess memory performance with fewer parameters.
RNN-W&D + LSTM + GRU	Combines LSTM and GRU layers sequentially within the deep path, creating a multi-timescale recurrent encoder for more expressive temporal feature extraction.

**Table 5 sensors-25-05222-t005:** Performance of ResRNN-Transformer under different time-step input strategies (3000-sample dataset).

Plan A				
	M1	M2	M3	M4
*R* ^2^	0.5476	0.6678	0.5733	0.5822
MAE	0.4615	0.3795	0.4850	0.3296
Plan B				
	M1	M2	M3	M4
*R* ^2^	0.5822	0.7359	0.6404	0.8562
MAE	0.4590	0.3515	0.4597	0.2839
Plan C				
	M1	M2	M3	M4
*R* ^2^	0.4888	0.7051	0.6264	0.8435
MAE	0.5258	0.3872	0.4620	0.2968
Plan D				
	M1	M2	M3	M4
*R* ^2^	0.7456	0.7579	0.6753	0.8522
MAE	0.3382	0.3316	0.4295	0.2789
Plan E				
	M1	M2	M3	M4
*R* ^2^	0.4101	0.2762	0.4272	0.8217
MAE	0.5437	0.6643	0.6047	0.3300

**Table 6 sensors-25-05222-t006:** ML model performance for modulus prediction across dataset sizes.

Performance Algorithms		M1			M2			M3			M4		
		MAE	MSE	*R* ^2^	MAE	MSE	*R* ^2^	MAE	MSE	*R* ^2^	MAE	MSE	*R* ^2^
Dataset: 3000													
ML	SVR	0.09	0.02	0.56	0.07	0.01	0.56	0.14	0.03	0.43	0.08	0.01	0.78
	Random Forest	**0.06**	**0.01**	**0.78**	**0.05**	0.01	0.80	**0.10**	**0.00**	0.67	**0.05**	**0.00**	**0.86**
	KNN	0.07	**0.01**	0.69	**0.05**	0.01	0.76	**0.10**	0.02	0.65	**0.05**	0.01	0.84
	Linear Regression	0.10	0.02	0.49	0.07	0.01	0.60	0.16	0.04	0.35	0.61	0.01	0.61
	Bayesian Ridge	0.10	0.02	0.49	0.07	0.01	0.60	0.16	0.04	0.35	0.09	0.01	0.61
	Gradient Boosting	**0.06**	**0.01**	0.77	**0.05**	**0.00**	**0.81**	**0.10**	0.02	**0.68**	**0.05**	0.01	**0.86**
	XGBoost	**0.06**	**0.01**	0.75	**0.05**	0.01	0.80	0.11	0.02	0.66	**0.05**	0.01	0.85
Dataset: 6000													
ML	SVR	0.06	0.01	0.94	0.07	0.01	0.22	0.06	0.01	0.82	0.08	0.01	−3.96
	Random Forest	**0.01**	**0.00**	**1.00**	**0.00**	**0.00**	0.94	**0.01**	**0.00**	0.97	**0.00**	**0.00**	**1.00**
	KNN	**0.01**	** 0.00 **	0.99	**0.00**	**0.00**	0.89	**0.01**	**0.00**	0.96	**0.00**	**0.00**	0.99
	Linear Regression	0.06	0.01	0.91	0.02	**0.00**	0.70	0.08	0.01	0.63	0.01	**0.00**	0.95
	Bayesian Ridge	0.06	0.01	0.91	0.02	**0.00**	0.70	0.08	0.01	0.63	0.01	**0.00**	0.95
	Gradient Boosting	**0.01**	**0.00**	**1.00**	**0.00**	**0.00**	**0.96**	**0.01**	**0.00**	**0.98**	**0.00**	**0.00**	**1.00**
	XGBoost	**0.01**	**0.00**	**1.00**	**0.00**	**0.00**	**0.96**	**0.01**	**0.00**	**0.98**	**0.00**	**0.00**	**1.00**
Dataset: 9000													
ML	SVR	0.06	**0.00**	0.95	0.06	0.01	0.35	0.06	0.01	0.83	0.08	0.01	−1.70
	Random Forest	**0.00**	**0.00**	**1.00**	**0.00**	**0.00**	0.98	**0.00**	**0.00**	**1.00**	**0.00**	**0.00**	**1.00**
	KNN	**0.00**	**0.00**	**1.00**	**0.00**	**0.00**	0.95	0.01	**0.00**	0.97	**0.00**	**0.00**	**1.00**
	Linear Regression	0.06	0.01	0.91	0.02	**0.00**	0.70	0.08	0.01	0.63	0.01	**0.00**	0.97
	Bayesian Ridge	0.06	0.01	0.91	0.02	**0.00**	0.70	0.08	0.01	0.63	0.01	**0.00**	0.97
	Gradient Boosting	0.01	**0.00**	**1.00**	**0.00**	**0.00**	0.99	0.01	**0.00**	0.98	**0.00**	**0.00**	**1.00**
	XGBoost	**0.01**	**0.00**	**1.00**	**0.00**	**0.00**	**0.99**	**0.01**	**0.00**	**0.98**	**0.00**	**0.00**	**1.00**

**Table 7 sensors-25-05222-t007:** DL model performance for modulus prediction across dataset sizes.

Performance Algorithms		M1			M2			M3			M4		
		MAE	MSE	*R* ^2^	MAE	MSE	*R* ^2^	MAE	MSE	*R* ^2^	MAE	MSE	*R* ^2^
Dataset: 3000													
DL	SimpleRNN	0.58	0.60	0.35	0.58	0.60	0.40	0.63	0.67	0.34	0.46	0.36	0.61
	ResRNN W&D	0.43	0.41	0.60	0.41	0.39	0.62	0.54	0.50	0.51	0.31	0.16	0.83
	ResRNN-W&D + LSTM	0.38	0.35	0.66	0.37	0.34	0.68	0.52	0.47	0.53	0.29	0.14	0.85
	ResRNN-W&D + GRU	0.40	0.38	0.62	0.39	0.36	0.66	0.51	0.47	0.54	0.30	0.15	0.85
	ResRNN-W&D + LSTM + GRU	**0.27**	**0.18**	**0.82**	**0.27**	**0.18**	**0.83**	**0.33**	**0.24**	**0.76**	**0.17**	**0.06**	**0.93**
Dataset: 6000													
DL	SimpleRNN	0.22	0.09	0.91	0.27	0.18	0.79	0.34	0.22	0.78	0.31	1.18	0.22
	ResRNN W&D	0.11	0.02	0.97	0.13	0.14	0.86	0.19	0.08	0.92	0.10	0.06	0.93
	ResRNN-W&D + LSTM	0.10	0.02	0.98	0.12	0.08	0.93	0.18	0.07	0.93	0.10	0.05	0.93
	ResRNN-W&D + GRU	0.06	0.01	**0.99**	0.09	0.03	0.97	0.14	0.05	0.95	0.13	0.04	0.96
	ResRNN-W&D + LSTM + GRU	**0.03**	**0.00**	**0.99**	**0.03**	**0.00**	**0.99**	**0.07**	**0.01**	**0.98**	**0.04**	**0.00**	**1**
Dataset: 9000													
DL	SimpleRNN	0.18	0.03	0.94	0.23	0.23	0.78	0.28	0.16	0.85	0.26	0.86	0.37
	ResRNN W&D	0.10	0.02	0.98	0.11	0.06	0.93	0.17	0.06	0.93	0.07	0.01	0.99
	ResRNN-W&D + LSTM	0.08	0.02	0.98	0.09	0.03	0.97	0.16	0.06	0.94	0.07	0.01	0.99
	ResRNN-W&D + GRU	0.05	0.01	0.99	0.08	0.03	0.97	0.11	0.04	0.96	0.05	0.01	0.96
	ResRNN-W&D + LSTM + GRU	**0.03**	**0.00**	**1.00**	**0.03**	**0.00**	**1**	**0.06**	**0.00**	**1**	**0.02**	**0.00**	**1**

**Table 8 sensors-25-05222-t008:** Average original pavement moduli in the dataset used for perturbation analysis.

M1	M2	M3	M4
2390.52	1629.64	78.42	53.02

**Table 9 sensors-25-05222-t009:** Model prediction summary for Gradient Boosting and XGBoost under Deflection-only and Plan D inputs.

Model	Modulus	Predicted Mean	Mean-Std	CI Width	Coverage Ratio
Deflection-only Inputs					
Gradient Boosting	M1	1743.86	126.56	496.11	0.30
	M2	1299.87	71.01	278.37	0.28
	M3	38.58	1.58	6.16	0.16
	M4	46.78	0.89	3.48	0.22
XGBoost	M1	1755.23	359.45	1409.04	0.86
	M2	1226.50	213.75	837.88	0.86
	M3	42.00	6.61	25.93	0.82
	M4	47.72	2.82	11.06	0.84
Plan D Inputs					
Gradient Boosting	M1	1668.70	41.84	164.03	0.12
	M2	1299.19	16.86	66.10	0.02
	M3	40.93	0.88	3.43	0.08
	M4	47.71	0.55	2.10	0.36
XGBoost	M1	1815.94	146.14	572.88	0.7
	M2	1213.08	110.13	431.73	0.73
	M3	44.87	3.20	12.56	0.80
	M4	48.15	1.33	5.20	0.73

**Table 10 sensors-25-05222-t010:** Model prediction summary for DL models under Deflection-only and Plan D inputs.

Model	Modulus	Predicted Mean	Mean-Std	CI Width	Coverage Ratio
Deflection-only Inputs					
ResRNN-W&D + GRU	M1	1570.26	432.02	1693.53	0.85
	M2	1467.11	188.17	737.64	0.56
	M3	35.22	5.86	22.97	0.50
	M4	47.40	5.01	19.64	0.72
ResRNN-W&D + GRU + LSTM	M1	1680.22	446.68	1751.00	0.84
	M2	1407.15	205.59	805.90	0.59
	M3	36.54	5.98	23.43	0.49
	M4	45.51	4.63	18.16	0.70
Plan D Inputs					
ResRNN-W&D + GRU	M1	1743.36	432.78	1696.48	0.79
	M2	1293.61	281.11	1827.31	0.86
	M3	38.77	10.52	41.26	0.75
	M4	45.68	7.37	28.81	0.85
ResRNN-W&D + GRU + LSTM	M1	1653.03	353.53	1385.83	0.76
	M2	1380.26	248.77	975.19	0.67
	M3	37.84	10.30	40.36	0.78
	M4	46.34	7.47	29.28	0.93

**Table 11 sensors-25-05222-t011:** Average original pavement moduli in the dataset used for perturbation analysis with weak subgrade conditions.

M1	M2	M3	M4
2390.52	1629.64	78.42	53.02

**Table 12 sensors-25-05222-t012:** Model prediction summary for ML models under Deflection-only and Plan D inputs with weak subgrade conditions.

Model	Modulus	Predicted Mean	Mean-Std	CI Width	Coverage Ratio
Deflection-only Inputs					
Gradient Boosting	M1	1628.77	156.17	612.20	0.27
	M2	1163.11	70.13	274.92	0.29
	M3	29.29	1.29	5.06	0.10
	M4	42.04	0.79	3.10	0.18
XGBoost	M1	1646.23	351.20	1376.71	0.98
	M2	1189.47	200.47	785.83	0.84
	M3	26.24	4.53	17.74	0.67
	M4	41.60	2.55	9.98	0.86
Plan D Inputs					
Gradient Boosting	M1	1578.17	48.74	191.08	0.08
	M2	1181.47	22.33	87.58	0.08
	M3	24.85	0.58	2.26	0.10
	M4	40.62	0.39	1.53	0.15
XGBoost	M1	1583.59	143.90	564.07	0.90
	M2	1195.19	88.51	346.96	0.73
	M3	22.05	2.49	9.78	0.65
	M4	41.23	0.89	3.49	0.69

**Table 13 sensors-25-05222-t013:** Model prediction summary for DL models under Deflection-only and Plan D inputs with weak subgrade conditions.

Model	Modulus	Predicted Mean	Mean-Std	CI Width	Coverage Ratio
Deflection-only Inputs					
ResRNN-W&D + GRU	M1	1495.30	448.43	1758.10	0.83
	M2	1119.50	233.76	916.33	0.72
	M3	29.08	6.63	26.00	0.64
	M4	43.08	4.81	20.81	0.75
ResRNN-W&D + GRU + LSTM	M1	1488.15	445.59	1722.10	0.81
	M2	1023.64	229.28	890.06	0.71
	M3	30.88	6.40	25.32	0.60
	M4	41.86	4.59	18.01	0.74
Plan D Inputs					
ResRNN-W&D + GRU	M1	1420.29	374.33	1473.94	0.78
	M2	1084.22	259.08	1105.58	0.73
	M3	29.26	9.71	38.05	0.47
	M4	41.99	6.84	26.96	0.59
ResRNN-W&D + GRU + LSTM	M1	1512.08	428.15	1579.00	0.86
	M2	1107.32	240.68	1182.30	0.70
	M3	26.11	9.31	36.51	0.44
	M4	41.95	6.70	26.26	0.60

## Data Availability

Data will be made available upon request.

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
