# Peer review of "Robust Pavement Modulus Prediction Using Time-Structured Deep Models and Perturbation-Based Evaluation on FWD Data"

_sensors, 2025, doi:10.3390/s25175222_

Round 1
Reviewer 1 Report
Comments and Suggestions for Authors
Please see the attachment below.

Author Response
Please see the attached PDF for our responses to the reviewers’ feedback.

Reviewer 2 Report
Comments and Suggestions for Authors
This study deals with the robust pavement modulus prediction using time-structured deep models and perturbation-based evaluation on Falling Weight Deflectometer data. The reviewer has some comments as follows:
1) Compared with the method in this study, other methods for determining the pavement modulus must be described. The advantages and disadvantages of each method must be discussed.
2) The authors need to explain why Gaussian noise was chosen in this study. Is Gaussian noise suitable for real conditions?
3) The study mentioned the experiment. Actual figures of the experiment should be added.
4) For section 3.4.1, each evaluation metric must be stated in terms of its meaning and how to use each metric to evaluate the results.
5) A detailed flowchart (step by step) must be added to help the reader understand the proposed method in the study.
6) For the results in tables 7-10, the error values between the predicted and actual modulus need to be added.
7) The citation of references is full of errors, for example [40,40], [Error! Reference source not found].
8) The manuscript should not have paragraphs that consist of only 2 to 3 sentences.
9) English needs to be polished.
Comments on the Quality of English LanguageEnglish needs to be polished.
Author Response

(The authors gave the same response as above.)

Round 2
Reviewer 1 Report
Comments and Suggestions for Authors
No more comments.
Reviewer 2 Report
Comments and Suggestions for Authors
The manuscript has been revised according to the reviewer's comments.